# The effect of *M. tuberculosis* lineage on clinical phenotype

**Duc Hong Du**[1], **Ronald B. Geskus**[1,2], **Yanlin Zhao**[3], **Luigi Ruffo Codecasa**[4], **Daniela Maria Cirillo**[5], **Reinout van Crevel**[2,6], **Dyshelly Nurkartika Pascapurnama**[7], **Lidya Chaidir**[8], **Stefan Niemann**[9,10], **Roland Diel**[11,12], **Shaheed Vally Omar**[13], **Louis Grandjean**[14], **Sakib Rokadiya**[14], **Arturo Torres Ortiz**[14], **Nguyễn Hữu Lân**[15], **Đặng Thị Minh Hà**[15], **E. Grace Smith**[16], **Esther Robinson**[16], **Martin Dedicoat**[16,17], **Le Thanh Hoang Nhat**[1], **Guy E. Thwaites**[1,2], **Le Hong Van**[1], **Nguyen Thuy Thuong Thuong**[1,2]⦿, **Timothy M. Walker**[1,2]⦿ *

1 Oxford University Clinical Research Unit, Ho Chi Minh City, Vietnam, 2 Centre for Tropical Medicine and Global Health, Nuffield Department of Medicine, University of Oxford, Oxford, United Kingdom, 3 CDC China, Beijing, China, 4 Regional TB Reference Centre/ Istituto Villa Marelli- ASST Grande Ospedale Metropolitano Niguarda, Milano, Italy, 5 IRCCS San Raffaele Scientific Institute, Milano, Italy, 6 Department of Internal Medicine and Radboud Center for Infectious Diseases, Radboud University Medical Center, Nijmegen, The Netherlands, 7 Research Center for Care and Control of Infectious Diseases, Universitas Padjadjaran, Sumedang, West Java, Indonesia, 8 Department of Biomedical Sciences, Faculty of Medicine, Universitas Padjadjaran, Sumedang, West Java, Indonesia, 9 Research Center Borstel, Borstel, Germany, 10 German Center for Infection Research, Partner Site Hamburg-Lübeck-Borstel-Riems, Germany, 11 University Hospital Schleswig-Holstein, Campus Kiel, Kiel, Germany, 12 Lung Clinic Grosshansdorf, Airway Disease Center North (ARCN), German Center for Lung Research (DZL), Grosshansdorf, Germany, 13 NICD, Johannesburg, South Africa, 14 University College London Hospital, London, United Kingdom, 15 Pham Ngoc Thach Hospital, Ho Chi Minh City, Vietnam, 16 TB Unit and National Mycobacterial Reference Service, UK Health Security Agency, Birmingham, United Kingdom, 17 University Hospitals Birmingham NHS Foundation Trust, Birmingham, United Kingdom

⦿ These authors contributed equally to this work.
* timothy.walker@ndm.ox.ac.uk

**Data Availability Statement:** All data used in the study are available in the supplementary materials. All code is available via the link supplied in the manuscript.

## Abstract

Six lineages of *Mycobacterium tuberculosis sensu stricto* (which excludes *M. africanum*) are described. Single-country or small observational data suggest differences in clinical phenotype between lineages. We present strain lineage and clinical phenotype data from 12,246 patients from 3 low-incidence and 5 high-incidence countries. We used multivariable logistic regression to explore the effect of lineage on site of disease and on cavities on chest radiography, given pulmonary TB; multivariable multinomial logistic regression to investigate types of extra-pulmonary TB, given lineage; and accelerated failure time and Cox proportional-hazards models to explore the effect of lineage on time to smear and culture-conversion. Mediation analyses quantified the direct effects of lineage on outcomes. Pulmonary disease was more likely among patients with lineage(L) 2, L3 or L4, than L1 (adjusted odds ratio (aOR) 1.79, (95% confidence interval 1.49–2.15), p<0.001; aOR = 1.40(1.09–1.79), p = 0.007; aOR = 2.04(1.65–2.53), p<0.001, respectively). Among patients with pulmonary TB, those with L1 had greater risk of cavities on chest radiography versus those with L2 (aOR = 0.69(0.57–0.83), p<0.001) and L4 strains (aOR = 0.73(0.59–0.90), p = 0.002). L1 strains were more likely to cause osteomyelitis among patients with extra-pulmonary TB, versus L2-4 (p = 0.033, p = 0.008 and p = 0.049 respectively). Patients with L1 strains showed

**Funding:** This research was funded in whole, or in part, by the Wellcome Trust [214560/Z/18/Z]. For the purpose of open access, the author has applied a CC BY public copyright licence to any Author Accepted Manuscript version arising from this submission. This work was supported by Wellcome Trust (grants 226007/Z/22/Z to LG) and the National Institute of Allergy and Infectious Diseases, National Institutes of Health (grant 1R01AI146338 to LG). Parts of this work have been funded by the Leibniz Science Campus Evolutionary Medicine of the LUNG (EvoLUNG), the Deutsche Forschungsgemeinschaft (DFG, German Research Foundation) under Germany's Excellence Strategy – EXC 2167 Precision Medicine in Inflammation as well as the Research Training Group 2501 TransEvo, and the German Center for Infection Reasearch (DZIF). TMW is a Wellcome Trust Clinical Career Development Fellow (214560/Z/18/Z). The funders played no role in the design, execution, or reporting of this study.

**Competing interests:** The authors have declared that no competing interests exist.

shorter time-to-sputum smear conversion than for L2. Causal mediation analysis showed the effect of lineage in each case was largely direct. The pattern of clinical phenotypes seen with L1 strains differed from modern lineages (L2-4). This has implications for clinical management and could influence clinical trial selection strategies.

## Introduction

*Mycobacterium tuberculosis* kills more people each year than any other pathogen besides SARS-CoV-2 [1]. Pulmonary disease is its most common form, but tuberculosis can disseminate throughout the host and manifest anywhere. Nine separate human adapted *Mycobacterium tuberculosis* lineages are described [2, 3]. Each lineage has emerged in its own geographical niche with some so-called specialist lineages having adapted to the host population with which they co-evolved, and other more generalist lineages having successfully spread throughout the world [4–6].

For the purpose of clinical trials, most diagnostics, and treatment, TB is still considered one disease rather than a family of different but related entities. However, data suggesting genuine differences exist are accumulating. The minimum inhibitory concentrations for pyrazinamide and pretomanid were recently found to be higher for lineage 1 than for other tested lineages [7, 8]. Epidemiological studies into the relationship between lineage and clinical phenotype have in the past indicated that lineage 1 in particular is more strongly associated with extrapulmonary disease than lineages 2–4 [9]. Of late an association and proposed mechanism have even been proposed for how lineage 1 is more likely to result in TB osteomyelitis than other lineages [10]. However, observational studies are fraught with potential confounders, such as immigration patterns, and only two small studies have so far simultaneously looked at data from both endemic countries and at countries where the majority of patients can be linked to another country of origin [11, 12].

After decades of limited progress, there are now a wealth of new drugs and even vaccines in the pipeline. Were differences between lineages thought to be major, this would need to be considered when designing clinical studies assessing new interventions. Here we present data from eight countries across four continents to explore the effect of *M. tuberculosis* lineage on, and risk factors for, clinical manifestations from the focus of disease to radiological markers of severity to treatment response.

## Methods

### Sample selection

We analysed existing datasets from three low-incidence countries where most patients with TB can be linked to another country of origin, and 5 high-incidence countries where immigration patterns are less relevant to local TB patterns. In each case data had originally been generated for other studies, independent of this one. Data from Germany and the UK are from metropolitan population studies, and include isolates from all patients for whom a culture was available and a lineage determinable over defined time periods. In Germany in particular this resulted in enrichment for pulmonary TB (90% vs. 71% background rate in the country [13]). Data from Italy were representative of all patients with pulmonary TB attending a clinic in Milan. Data from China were from a national pulmonary TB prevalence study, and those from South Africa were from a surveillance study of bedaquiline resistance among patients with rifampicin resistant pulmonary TB. Data from Peru are from a locally representative

**Table 1. Descriptions of datasets.**

| Country | Data set description | Number of patients / isolates | Time interval | Bias |
|---|---|---|---|---|
| Germany | Population study of TB in Hamburg | 673 | 2008–2017 | Enriched for pulmonary TB due to ease of sampling |
| UK | Population study of TB in Birmingham | 1653 | 2009–2019 | |
| Italy | All pulmonary TB isolates from patients in Milan | 90 | 2018–2019 | Pulmonary TB only |
| South Africa | Bedaquiline resistance prevalence study among patients with MDR-TB | 175 | 2015–2019 | Enriched for rifampicin resistant; pulmonary TB only |
| Vietnam 1 | Observational studies of all pulmonary TB | 1536 | 2008–2011 | Pulmonary TB only |
| Vietnam 2 | Observational study of MDR pulmonary TB | 257 | 2017–2020 | Pulmonary MDR-TB only |
| Vietnam 3 | Observational study of rifampicin susceptible pulmonary TB | 537 | 2017–2020 | Pulmonary Rifampicin-susceptible TB only |
| Vietnam 4 | Clinical trial of TB meningitis | 370 | 2011–2014 | TB meningitis only |
| Vietnam 5 | Clinical trial of TB meningitis in HIV infected individuals | 111 | 2017–2020 | TB meningitis only; HIV+ |
| Vietnam 6 | Clinical trial of TB meningitis in HIV uninfected individuals | 105 | 2018–2020 | TB meningitis only; HIV- |
| Indonesia 1 | Clinical trial of TB meningitis | 106 | 2006–2016 | TB meningitis only |
| Indonesia 2 | Observational study of pulmonary TB, enriched for MDR-TB | 765 | 2006–2016 | Pulmonary TB only; enriched for MDR-TB |
| China | Population prevalence study of pulmonary TB | 5445 | 2013–2017 | Pulmonary TB only |
| Peru | Observational study of pulmonary TB | 429 | 2018–2019 | Pulmonary TB only |

observational study on pulmonary TB. Data from Vietnam and Indonesia originated both from observational studies on pulmonary TB, and from clinical trials on TB meningitis. See Table 1 for details.

## Variables

All isolates had been assigned a lineage based on their whole-genome sequence before being shared for the purpose of this analysis. No further WGS analysis was conducted here. Outcome variables included locus of disease, cavities on chest radiographs (CXR), Timika scores (the higher the score, the more severe the disease on CXR) [14], and time to smear and culture conversion. These were all obtained from local clinicians, radiologists and laboratories. We included other variables plausibly effecting outcome, including sex, diabetes, HIV infection, age, isoniazid or rifampicin resistance, country of origin of the data and whether the patient was born in that same country or not. S1 Table shows which datasets contained which variables.

## Statistical analysis

In order to understand which variables should be included as potential confounders, we developed Directed Acyclic Graphs (DAGs) identifying the hypothesized causal relationships between individual variables (S1A–S1C Fig). Logistic regression was used to relate the exposure (lineage) to (A) pulmonary vs. extra-pulmonary TB, and (B) the presence of pulmonary cavities in patients with pulmonary TB. In each case age, country of data origin, birth in that country, diabetes and HIV infection were included in the model where these variables were available. As not all datasets included all the same variables, we first analysed each country separately to assess risk factors for the outcome, before performing a causal analysis after pooling data from a subset of countries adjusting for covariates. Confidence intervals and p-values were computed based on the likelihood ratio test and result reported with Firth's correction

where numbers were small. Age was included in the models using restricted cubic splines with three knots (the knots were chosen at the 10%, 50% and 90% percentile of the values of the variable) to allow for potential non-linear relationships where applicable. Linear regression was used to examine the effect of lineage on Timika CXR score. As the score had a skewed distribution, we used the Box-Cox procedure to find a suitable transformation (i.e. a square root transformation) that makes the outcome variable more symmetrically distributed. Multivariable multinomial logistic regression was used to analyse the association between lineage and different forms of extra-pulmonary TB with a Wald test to obtain confidence intervals and p-values.

We investigated the effect of lineage on time to smear and culture conversion using an accelerated failure time model under the assumption of a log-logistic baseline distribution based on the Akaike information criterion (AIC) of different parametric fitted models including log-logistic (lowest AIC), exponential, Weibull, gamma and log-normal baseline distribution. The nature of the time-to-event data varied by study so these had to be standardised to achieve a lower and upper bound for the window of time in which smear or culture conversion is likely to have occurred. Where only one date–that of a first negative sample–was available, this was set as the upper bound of the interval and the lower bound set according to the expected monthly sampling date prior to that (day 0, 30, 60, 90 etc.). Where results from multiple sampling dates were available for a patient, the upper bound was defined as the first of the first two consecutive negatives, and the last preceding positive set as the lower bound. Where the first negative sample was also the last sample documented, this was treated as the conversion date because we assume that there were more samples but they were all negative and therefore not reported. See appendix for further details. We analysed the same data using a Cox proportional-hazards model allowing for interval censored data to explore whether the findings were consistent.

We conducted causal mediation analyses to investigate the key drivers of our selected outcomes. To this end we focussed on pathways of effects from lineage to outcome, quantifying the effect-size attributable to mediators, rather than just assessing the marginal effect of lineage by standard multivariate models alone. Intermediary and confounding variables were predetermined from the DAGs (S1A–S1C Fig). Effects of lineage on outcomes were divided into natural direct and indirect effects via the regression-based approach with closed-form parameter function estimation or direct counterfactual imputation estimation or via the g-formula approach where applicable [15, 16].

Direct counterfactual imputation estimation was conducted by fitting a model for the outcome conditional on exposure (*A*), mediator (*M*) and covariates (*C*) using the original dataset, *E(Y|A,M,C)*. We fitted a multinomial regression model for the distribution of the mediator, conditional on exposure and covariates, and simulated the counterfactuals of the mediator for each value of the exposure. We then imputed the counterfactuals of expected outcome for each individual observation conditional on exposure, imputed mediator values and covariates. The average over the individuals was taken to obtain the counterfactual estimate, and causal -, direct -, indirect -, and total-effects were calculated.

Exposure-mediator interaction was considered. We also assessed the contribution of the effect of lineage on outcome as operating through mediators, using the "proportion mediated" where applicable, i.e. when direct and indirect effects are in the same direction [17]. The proportion mediated is defined as the ratio of the natural indirect effect (NIE) to the total effect (TE) (on the logit scale), that is,

$$PM = \frac{NIE}{TE}$$

With binary outcomes, the proportion mediated can be calculated from the natural direct effects odds ratio $OR^{NDE}$ and the natural indirect effect odds ratio $OR^{NIE}$ when the outcome is rare and a logistic model was used [18]:

$$\frac{OR^{NDE}\,(OR^{NIE} - 1)}{(OR^{NDE} \times OR^{NIE} - 1)}$$

When the outcome is not rare, the relative risk or rate ratio (RR) from a log-linear model was reported instead of an OR to prevent bias if implemented via the regression-based approach with closed-form parameter function estimation. Using the direct counterfactual imputation estimation will ignore this problem [17].

We reported estimated effects (ORs and RRs) for both natural direct and indirect effects with 95% CIs and p-values obtained by two methods: delta and bootstrapping [17]. We reported estimated ORs using the direct counterfactual imputation estimation and 95%CIs and p-values obtained by bootstrapping method as the main results.

Analyses were performed in R version 4.2.0, and the packages 'ggplot2', 'icenReg', and 'CMAverse' [19–22].

### Ethics and data statements

IRB approvals were obtained from the University of Lübeck (Germany); Universidad Peruana Cayetano Heredia via the Peruvian Ministry of Health (ref. 100252) (Peru); Pham Ngoc Thach Hospital Ho Chi Minh City and the University of Oxford (OxTREC) (Vietnam); China CDC (China); University of Witwatersrand Human Research Ethics Committee (ref. M160667) (South Africa); Hasan Sadikin Hospital/Faculty of Medicine of Universitas Padjadjaran (clinical trial ref. NCT02169882) (Indonesia); Observational/Interventions Research Ethics Committee, London School of Hygiene and Tropical Medicine (ref: 6449) (Indonesia); institutional review boards in Indonesia in the context of the TANDEM study (Indonesia); Ospedale San Raffaele, Milan (ref: OSR 82/DG 26/2/10, amended 11/12/14) (Italy). Data from the UK data were obtained entirely from the published literature [23]. Those data had been collected under public health law with no need for further IRB approval or individual patient consent.

All data are available in S19 Table. R code is available here: https://github.com/duhongduc/lineage. Raw WGS data are not provided as were not part of this analysis.

## Results

Data on 12,547 patients were obtained from eight countries. Lineage 1 accounted for 1,024 (8%), lineage 2 for 6,477 (52%), lineage 3 for 796 (6%) and lineage 4 for 3,955 (32%) observations. Data on lineage were missing for 254 patients, 40 labelled as a mixture of lineages were excluded, and 7 represented other lineages (5 *M. africanum* from Germany, 1 from Italy, and 1 *M. bovis* from Peru) that were too rare to justify inclusion in this analysis. Table 1 and S1 Table show the numbers of patients and isolates from each country, and how these were sampled. The largest collections came from China, Vietnam, and the UK, with each of the other countries contributing data from fewer than 1,000 patients.

### Pulmonary vs. extra-pulmonary disease

Datasets from Vietnam, Indonesia, Germany and the UK contained data on patients with pulmonary TB and on patients with extra-pulmonary TB so could be used to assess whether lineage influences the spread of TB beyond the lung. For this purpose, patients known to have both pulmonary and extra-pulmonary TB were classified as 'extra-pulmonary TB'.

Data on patient sex were available for all countries, and on patient age and HIV infection from Germany, Indonesia and Vietnam. However, as data on HIV from Vietnam were from two TB meningitis clinical trials on HIV infected and uninfected individuals respectively, this variable was excluded from Vietnam to avoid bias. Data on diabetes were available from Germany and Indonesia only.

We first analysed each country individually to explore risk factors for each outcome. In Germany and Vietnam lineages 2 and 4 were more likely than lineage 1 to associate with pulmonary TB after controlling for immigration, and in the case of Germany, HIV infection and diabetes as well (aOR = 4.88 (95% CI 1.41–19.8), p = 0.016 and aOR = 3.16 (1.29–7.23), p = 0.008 respectively for Germany; aOR = 1.7 (1.38–2.09), p<0.001 and aOR = 2.43 (1.66–3.63), p<0.001 respectively for Vietnam). No difference was seen by lineage for Indonesia where the trend was in the opposite direction. Increasing age was a risk factor for pulmonary TB in Vietnam and Indonesia (S2 Table).

We next performed a causal analysis of each country controlling for immigration, and age where available. In Germany, Vietnam and Indonesia we saw the same patterns as in the analysis of risk factors. A similar pattern was also seen in the UK where lineages 2, 3 and 4 were all more likely to cause pulmonary TB as compared to lineage 1 (aOR = 2.21 (1.31–3.81), p = 0.003; aOR = 1.51 (1.11–2.07), p = 0.009; aOR = 2.19 (1.59–3.04), p<0.001, respectively). To combine data from the 4 countries and still control for age, we needed to impute age for the UK patients. We based this on the mean age in Germany, which a sensitivity analysis showed did not bias the results. In the consequent combined analysis, lineages 2, 3 and 4 were again all more likely to cause pulmonary TB than lineage 1 (aOR = 1.79 (1.49–2.15), p<0.001; aOR = 1.40 (1.09–1.79), p = 0.007; aOR = 2.04 (1.65–2.53), p<0.001 respectively) (Fig 1A, S3 Table).

A causal mediation analysis adjusting for country (Germany, Indonesia and Vietnam), immigration, and age indicated that the effect of lineage on pulmonary TB is largely independent (direct effect) but that some of the effect is also mediated through drug resistance. Lineage 2 had a higher probability of drug resistance than lineage 1, and drug resistance in turn led to a higher probability of pulmonary TB (Fig 1B, S4 Table).

## Pulmonary cavity vs. no cavity

Given the association between lineages 2, 3 and 4 and pulmonary disease, we explored whether these lineages are also more likely than lineage 1 to lead to lung cavities as seen on CXR. Cavities are a marker of severity of disease, and are considered a risk factor for onward transmission [24]. Data on the presence of cavities were available from Germany, Italy, Vietnam, Indonesia, China and Peru, and any missing data was understood to be missing completely at random.

Logistic regression was used to assess each country's data for risk factors for cavity formation. Diabetes was a risk factor for cavities on CXR in China and Peru (aOR = 1.27 (1.02–1.56), p = 0.028 and aOR = 4.04 (1.37–17.3), p = 0.026 respectively, S7 Table). As age and immigration were identified as potential confounders in the DAG, further analyses were restricted to include only these co-variables.

In Vietnam, where 490 lineage 1 isolates were present, both lineages 2 and 4 were found to be less likely to associate with cavity formation than lineage 1 (aOR = 0.66 (0.53–0.81), p<0.001; aOR = 0.47 (0.33–0.67), p<0.001, respectively). No difference was seen between lineages for data from Germany, Italy, Indonesia, China or Peru (S6 Table). However, the maximum number of lineage 1 isolates in any one of these countries was just 29. In Peru there were zero lineage 1 isolates, leaving L2 as the reference. For the causal analysis we pooled the data

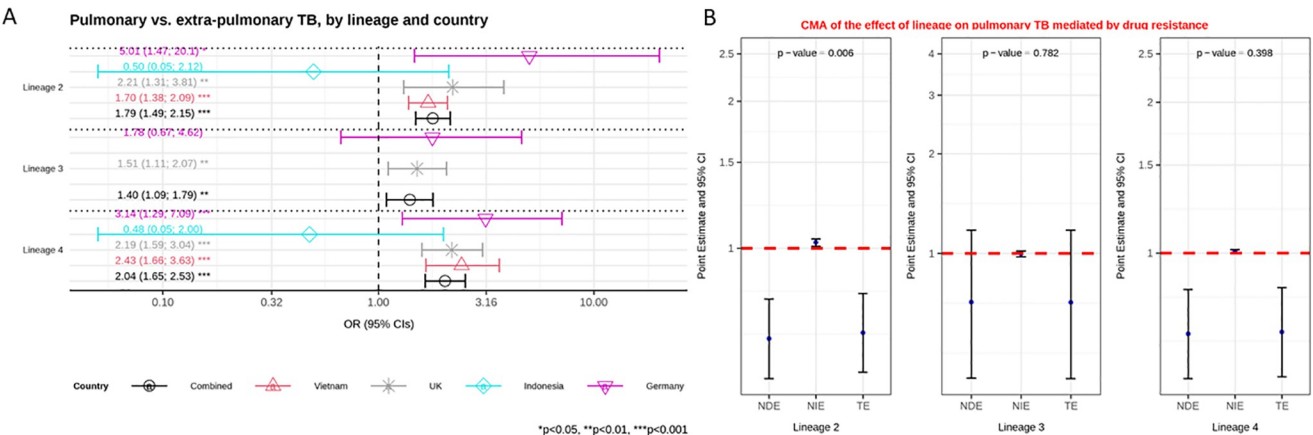

**Fig 1. A**: Multivariable logistic regression model on the association between lineage and pulmonary versus extra-pulmonary tuberculosis (TB) controlling for age and immigration. Estimated odd ratios (ORs) and bars representing 95% confidence intervals (CIs) are shown on the x-axis for lineage 2, 3 and 4, compared to lineage 1 as reference, for each country as well as for all these countries combined. P-values denote evidence of the associations of lineage and pulmonary TB. **B**: Causal mediation analysis (CMA) on the effect of lineage on pulmonary TB, mediated by drug resistance. Estimated odds ratio (ORs) and bars representing 95% confidence intervals (CIs) are shown on the y-axis for each decomposition effect including NDE: natural direct effect odds ratio; NIE: natural indirect effect odds ratio; and TE: total effect odds ratio for lineage 2, lineage 3, and lineage 4, compared to lineage 1 as reference. All multivariable models adjusted for country, immigration, and age are shown. P-values denote evidence of natural indirect effect of lineage on pulmonary TB mediated through drug resistance. The red horizontal lines indicate the thresholds of the results (ORs) of interest.

from all five countries. Controlling for age, immigration and country we found that lineage 1 was more likely to cause cavities than lineage 2 and 4. The odds ratio for lineage 3 was similar to the other two modern lineages but the confidence intervals allowed for the possibility that it might behave similar to lineage 1 (Fig 2A, S6 Table).

A causal mediation analysis was performed for Germany, Italy, Indonesia, China and Vietnam as there was data on drug resistance for these. This indicated a protective natural direct

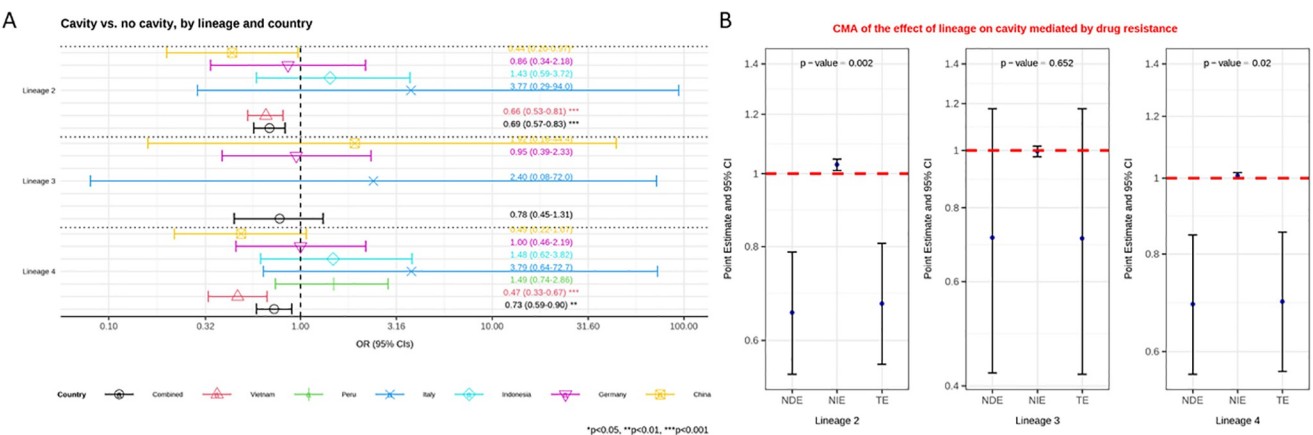

**Fig 2. A**: Multivariable logistic regression model on the association between lineage and the presence of cavity versus no cavity among patients with pulmonary TB, controlling for age and immigration. Estimated odd ratios (ORs) and bars representing 95% confidence intervals (CIs) are shown on the x-axis for lineage 2, 3 and 4, compared to lineage 1 as reference, for each country as well as for all these countries combined. P-values denote evidence of the associations of lineage and cavity. **B**: Causal mediation analysis (CMA) on the effect of lineage on cavity, mediated by drug resistance. Estimated odds ratio (ORs) and bars representing 95% confidence intervals (CIs) are shown on the y-axis for each of the decomposition effect including NDE: natural direct effect odds ratio; NIE: natural indirect effect odds ratio; and TE: total effect odds ratio of lineage 2, lineage 3, and lineage 4, compared to lineage 1 as reference. All multivariable models adjusted for country, immigration, and age are shown. P-values denote evidence of natural indirect effect of lineage on cavity mediated through drug resistance. The red horizontal lines indicate the thresholds of the results (ORs) of interest.

effect of lineage from cavity formation for lineages 2 and 4, compared to lineage 1. Here however the natural indirect effect of drug resistance increased the odds of cavity for lineage 2 over lineage 1. Lineage 2 has higher odds of drug resistance versus lineage 1, and drug resistance in turn leads to higher odds of cavity formation (Fig 2B, S8 Table).

Severity of diseases on CXR had been assessed for data from Peru, Italy, Indonesia and Vietnam using the Timika scoring system to which the presence of cavities on the CXR contributes [14]. As the score had a skewed distribution, we used the Box-Cox procedure to find a suitable transformation (a square root transformation) to generate a more symmetrical distribution. Although lineage 1 was associated with a higher Timika score than lineage 4 in Vietnam, this effect was not observed elsewhere (S10 Table).

## Extra-pulmonary disease

Having found that in these data lineage 1 was associated with extra-pulmonary TB relative to other lineages, we investigated whether there was a particular focus of extra-pulmonary diseases which lineage 1 favours. The population-based data from Germany and the UK were most suited to assess this. Data from the UK reported pulmonary TB; TB meningitis; TB osteomyelitis; or other types of TB. Data from Germany were more detailed on other forms of TB, including lymph node and pleural disease, although there were only 3 patients with TB meningitis and 10 with osteomyelitis across all 4 lineages (S1 Table). Data from the two countries were therefore pooled, coding the sites of disease for Germany as for the UK. One case with two sites of extra-pulmonary TB was dropped from this analysis. Using multivariable multinomial logistic regression we took pulmonary TB as the reference and assessed the odds ratio of TB meningitis, osteomyelitis or other forms of extra-pulmonary TB, given lineage. TB osteomyelitis was significantly more likely than other non-meningeal foci of extra-pulmonary TB for lineage 1 as compared to lineages 2, 3 and 4 (p = 0.032, p = 0.01, and p = 0.049 respectively; S11 and S12 Tables; Fig 3 and S2 Fig).

## Time to smear and culture conversion

To explore a different manifestation of the clinical phenotype by lineage, we related lineage to the time to sputum smear and culture conversion. Data from Indonesia, Italy and Vietnam were available to model time to smear and culture conversion whilst correcting for country, immigration, and age as potential confounders.

In the accelerated failure time (AFT) model, time to both smear and culture conversion increased with lineage 2 compared to lineage 1 (Fig 4A; S13 and S14 Tables). The Cox proportional-hazards model produced similar results, showing that the 'hazards' of having culture or smear conversion at any given time was lower for lineages 2 and 4 than for lineage 1 (i.e. longer time to conversion), after adjusting for age, country and immigration (S3 Fig; S17 and S18 Tables).

Causal mediation analysis looking at cavity and drug resistance showed no evidence of a natural indirect effect mediated through drug resistance and cavity on time to culture conversion (Fig 4B, S15 Table). We found moderate evidence that lineage 4 shortened time to smear conversion, compared to lineage 1, as a natural indirect effect mediated through drug resistance and cavity, in the same direction as the natural direct effect (S4 Fig, S16 Table).

## Discussion

We explore causal relationship between *M. tuberculosis* lineage and TB clinical phenotypes using a large dataset derived from 3 low-incidence and 5 high-incidence countries. In these data lineage 1 is a risk factor for extra-pulmonary TB, with a suggestion that it favours TB

**Fig 3.** Multinomial, multivariable regression for pulmonary tuberculosis (TB) vs. three types of extra-pulmonary TB (TB meningitis, TB osteomyelitis, and other forms of extra-pulmonary TB), by lineage, controlling for country and immigration. Estimated odd ratios (ORs) and bars representing 95% confidence intervals (CIs) are shown on the x-axis for the odds ratios of being lineage 2, 3 and 4, compared to lineage 1 as reference, comparing pulmonary TB to each of form of extra-pulmonary TB ("TBM"—TB meningitis, "Osteo"—TB osteomyelitis, and "Other"—other forms of extra-pulmonary TB) on the y-axis. P-values denote evidence of the association between lineages and different forms of extra-pulmonary TB compared to pulmonary TB.

osteomyelitis in particular. When lineage 1 does cause pulmonary TB, we find that it is a risk factor for pulmonary cavities.

Lineage 1 is ancestral to the other 3 lineages assessed in this study [25]. It is most prevalent in south Asia and has been linked to both extra-pulmonary TB and to more severe disease phenotypes [9, 26, 27]. It is plausible that the modern lineages (2, 3 and 4) are more adapted to manifesting as pulmonary TB, thereby favouring more transmission [10]. This would have implications for the epidemiology of multi-drug resistant TB given that this may be less common in lineage 1 [28]. Indeed, there is evidence from Vietnam and elsewhere that lineage 1 is being out-competed by the other three global lineages [29, 30]. Our finding that lineage 2 has a longer time to smear and culture conversion is consistent with this [11, 31], although we find no difference between lineages 1 and 4. It may be that if lineage 1 does have a preponderance to cause extra-pulmonary TB, that it compensates by forming cavities when it does manifest as pulmonary TB, boosting opportunities for onward transmission when it can [24]. We however know of few other data to support this hypothesis.

The results of the causal mediation analyses indicate that the effect of lineage on locus of disease and cavity formation is only minimally mediated through drug resistance. That some effect is mediated through cavity is an intuitive finding given that drug resistance could lead to

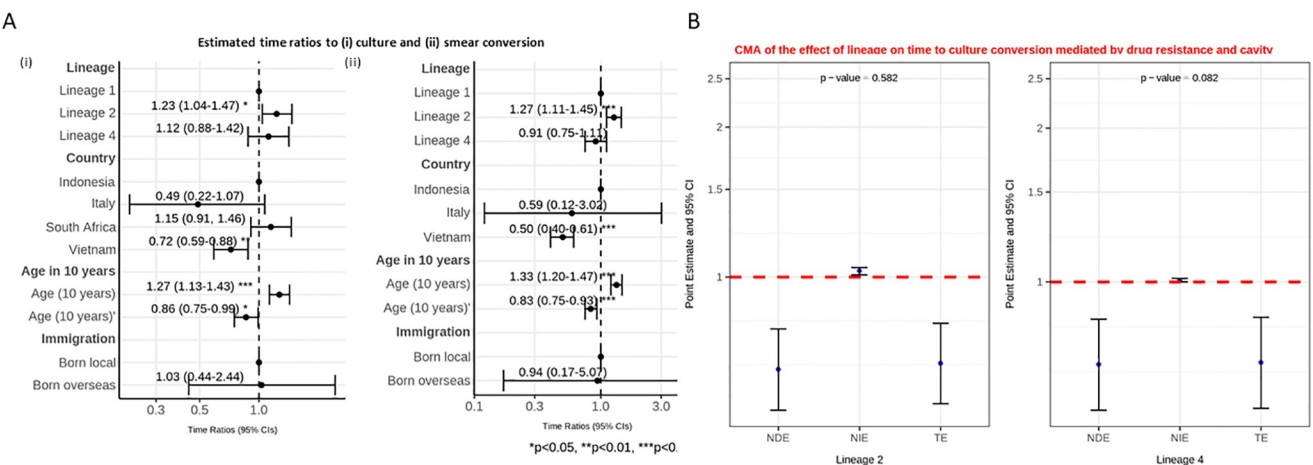

**Fig 4. A**: Interval censored regression using accelerated failure time models on the association between lineage and time to culture (i) and smear (ii) conversion controlling for age, country and immigration status. Estimated time ratios and bars representing 95% confidence intervals (CIs) are shown on the x-axis. Data from Indonesia, Italy and South Africa all had interval censored data whereas the data from Vietnam were binary (< = 60 days or >60 days). The Vietnamese data were therefore converted to interval data ("0 to 60" if < = 60; and "61 to ∞" if >60). P-values denote evidence of the associations of lineage and time to culture or smear conversion. **B**: Causal mediation analysis (CMA) on the effect of lineage on time to culture conversion mediated by drug resistance and cavity. Estimated time ratios and bars representing 95% confidence intervals (CIs) are shown on the y-axis for each of the decomposition effect including NDE: natural direct effect odds ratio; NIE: natural indirect effect odds ratio; and TE: total effect odds ratio of lineage 2 and lineage 4, compared to lineage 1 as reference. All multivariable models adjusted for country, immigration, and age are shown. P-values denote evidence of natural indirect effect of lineage on time to culture conversion mediated through drug resistance and cavity. The red horizontal lines indicate the thread holds of the results (ORs) of interest.

worse disease where it is more difficult to treat. Although we might have expected to find some effect on time to smear and culture conversion mediated through drug resistance and cavity, this was only observed for time to smear, but not for time to culture conversion.

Results from observational studies on the association between *M. tuberculosis* lineage and clinical phenotype must be treated with caution. Other potential confounders, include host susceptibility, co-morbidities, and local socio-economic circumstances. Most previous studies that we are aware of have focussed only on data from single countries [9], and those that have focussed on more than one country have been small [11, 12]. We have gathered data from 8 countries and over 12,000 patients. Our data represent both endemic settings and countries where the majority of TB is most likely acquired elsewhere. Although we could not control for all potential confounders, controlling for country of data origin may have captured at least some. Indeed, the recent identification of a potential mechanism for the association between lineage 1 and TB osteomyelitis provides support for our approach and findings [10].

Limitations of our study are that our data were derived from a wide range of studies, each primarily designed for another purpose. Our sampling frames ranged from population prevalence studies to clinical trials, and from studies that included all clinical samples over defined periods of time and space to studies that selected only for one manifestation of disease. These intrinsic biases need to be recognised when constructing each of the presented models. However, none of the studies we pooled data from selected on the basis of lineage as such data could not be known until after the isolates had been analysed. As such, the distribution of lineages within each collection should reflect the lineages' natural tendency to result in the outcomes in question, and not any extrinsic selection bias.

Other limitations include heterogeneity of measurement for individual variables. For example, cavities were reported from CXRs by attending physicians or by radiologists in the different countries without centralised, standardised reporting across the study. Time to

smear and culture conversion was reported differently for different studies, with some data sets requiring more assumptions to define time intervals than others. We had no data on second generation immigrants from high to low-burden countries so were unable to control for this potential confounder. Also, no data on patient treatment were available to control for whether they were on effective regimens. As all data were gathered from pre-existing studies, there was no opportunity to remedy this. Despite the inevitable noise in our data, the large size of the combined data set will have helped overcome a degree of random error by increasing power.

As our understanding of the phenotypic differences between *M. tuberculosis* lineages increases, we need to keep re-assessing whether to factor differences into study designs [11]. Increased MICs for pyrazinamide and pretomanid for lineage 1 are clearly relevant to clinical trials of those drugs. Whether different clinical phenotypes should be considered as well remains an open question, but it could be of interest if certain lineages are more prone to cavity formation or different durations to smear or culture conversion as these variables could plausibly impact outcome. Multi-centre, multi-country clinical studies are usually designed to capture diversity across host populations, but it may be that incorporating the full genetic diversity of the pathogen in such studies is equally important.

## Supporting information

**S1 Fig. Directed Acyclic Graph (DAG) on the causal assumptions underlying the effect of Lineage on (A) Pulmonary tuberculosis (TB); (B) the presence of Cavity; and (C) Time to culture/smear conversion.** Arrows indicate the direction of the effect. Exposure, Mediator, Outcome and Confounders listed below each graph.
(TIF)

**S2 Fig. Predicted probability for pulmonary tuberculosis (TB) vs. three types of extra-pulmonary TB (TB meningitis, TB osteomyelitis, and other forms of extra-pulmonary TB), by lineage "lin1234" (Lineage 1, 2, 3, and 4), country (Germany and UK) and immigration ("immi" (0 –born local and 1 –born overseas) from multinomial, multivariable regression.**
(TIF)

**S3 Fig. Interval censored regression using proportional hazards models on the association between lineage and time to culture (A) and smear (B) conversion controlling for age, country and immigration.** Estimated hazards ratios and bars representing 95% confidence intervals (CIs) are shown on the x-axis. Data from Indonesia, Italy and South Africa all had interval censored data whereas the data from Vietnam were binary ($< = 60$ days or $>60$ days). The Vietnamese data were therefore converted to interval data ("0 to 60" if $< = 60$; and "61 to $\infty$" if $>60$). P-values denote evidence of the associations of lineage and time to culture or smear conversion.
(TIF)

**S4 Fig. Causal mediation analysis (CMA) on the effect of lineage on time to smear conversion mediated by drug resistance and cavity.** Estimated time ratios and bars representing 95% confidence intervals (CIs) are shown on the y-axis for each of the decomposition effect including NDE: natural direct effect odds ratio; NIE: natural indirect effect odds ratio; and TE: total effect odds ratio of lineage 2 and lineage 4, compared to lineage 1 as reference. All multivariable models adjusted for country, immigration, and age are shown. P-values denote evidence of natural indirect effect of lineage on time to smear conversion mediated through drug resistance and cavity. The red horizontal lines indicate the thread holds of the results (ORs) of

interest.
(TIF)

**S1 Table. Variables that were available from each country's dataset, listed by lineage (L1, L2, L3 and L4).**
(XLSX)

**S2 Table. Multivariable logistic regression on the association between lineage and pulmonary tuberculosis (TB) versus extra- pulmonary TB, controlling for age, diabetes, HIV infection, and immigration status.**
(XLSX)

**S3 Table. Multivariable logistic regression on the association between lineage and pulmonary tuberculosis (TB) versus extra- pulmonary TB, controlling for age, and immigration status.**
(XLSX)

**S4 Table. Causal mediation analysis on the association between lineage and pulmonary tuberculosis (TB) versus extra- pulmonary TB, mediated by drug resistance, controlling for country, age, and immigration status.**
(XLSX)

**S5 Table. Causal mediation analysis on the association between lineage and pulmonary tuberculosis (TB) versus extra- pulmonary TB, mediated by drug resistance, controlling for country, age, diabetes, and immigration status.**
(XLSX)

**S6 Table. Multivariable logistic regression on the association between lineage and the presence of cavity on chest radiograph, controlling for age, and immigration status.**
(XLSX)

**S7 Table. Multivariable logistic regression on the association between lineage and the presence of cavity on chest radiograph, controlling for age, diabetes, HIV infection, and immigration status.**
(XLSX)

**S8 Table. Causal mediation analysis on the association between lineage and the presence of cavity on chest radiograph, mediated by drug resistance, controlling for country, age, and immigration status.**
(XLSX)

**S9 Table. Causal mediation analysis on the association between lineage and the presence of cavity on chest radiograph, mediated by drug resistance, controlling for country, age, diabetes, and immigration status.**
(XLSX)

**S10 Table. Multivariable linear regression on the association between lineage and Timika CXR score, controlling for age, and immigration status.**
(XLSX)

**S11 Table. Multivariable multinomial logistic regression on the association between lineage and different forms of extra-pulmonary tuberculosis (TB), controlling for immigration status and country of data origin.**
(XLSX)

**S12 Table. Wald test results from multivariable multinomial logistic regression on the association between lineage and different forms of extra-pulmonary tuberculosis (TB), controlling for immigration status and country of data origin.**
(XLSX)

**S13 Table. Multivariable accelerated failure time model on the association between lineage and time to culture conversion, controlling for age, country, and immigration status.**
(XLSX)

**S14 Table. Multivariable accelerated failure time model on the association between lineage and time to smear conversion, controlling for age, country, and immigration status.**
(XLSX)

**S15 Table. Causal mediation analysis on the association between lineage and time to culture conversion, mediated by drug resistance and cavity, controlling for country, age, and immigration status.**
(XLSX)

**S16 Table. Causal mediation analysis on the association between lineage and time to smear conversion, mediated by drug resistance and cavity, controlling for country, age, diabetes, and immigration status.**
(XLSX)

**S17 Table. Multivariable Cox proportional-hazards model on the association between lineage and time to culture conversion, controlling for age, country, and immigration status.**
(XLSX)

**S18 Table. Multivariable Cox proportional-hazards model on the association between lineage and time to smear conversion, controlling for age, country, and immigration status.**
(XLSX)

**S19 Table. Dataset.**
(XLSX)

## Acknowledgments

We would like to acknowledge the help of Dr. Stefania Torri of the Regional TB Reference Laboratory, Grande Ospedale Metropolitaqno Niguarda, Milan Italy, and of Dr. Simone Villa, of the Centre for Multidisciplinary Research in Health Science University of Milan, Italy.

## Author Contributions

**Conceptualization:** Guy E. Thwaites, Nguyen Thuy Thuong Thuong, Timothy M. Walker.

**Data curation:** Yanlin Zhao, Luigi Ruffo Codecasa, Daniela Maria Cirillo, Reinout van Crevel, Dyshelly Nurkartika Pascapurnama, Lidya Chaidir, Stefan Niemann, Roland Diel, Shaheed Vally Omar, Louis Grandjean, Sakib Rokadiya, Arturo Torres Ortitz, E. Grace Smith, Esther Robinson, Martin Dedicoat, Le Hong Van, Nguyen Thuy Thuong Thuong, Timothy M. Walker.

**Formal analysis:** Duc Hong Du, Ronald B. Geskus, Nguyen Thuy Thuong Thuong, Timothy M. Walker.

**Investigation:** Ronald B. Geskus, Timothy M. Walker.

**Methodology:** Duc Hong Du, Ronald B. Geskus, Le Thanh Hoang Nhat, Nguyen Thuy Thuong Thuong, Timothy M. Walker.

**Project administration:** Nguyễn Hữu Lân, Đặng Thị Minh Hà, Nguyen Thuy Thuong Thuong, Timothy M. Walker.

**Resources:** Nguyễn Hữu Lân, Đặng Thị Minh Hà, Guy E. Thwaites.

**Software:** Duc Hong Du, Ronald B. Geskus, Timothy M. Walker.

**Supervision:** Ronald B. Geskus, Guy E. Thwaites, Nguyen Thuy Thuong Thuong, Timothy M. Walker.

**Validation:** Duc Hong Du, Ronald B. Geskus, Timothy M. Walker.

**Visualization:** Duc Hong Du, Ronald B. Geskus, Timothy M. Walker.

**Writing – original draft:** Timothy M. Walker.

**Writing – review & editing:** Duc Hong Du, Ronald B. Geskus, Yanlin Zhao, Luigi Ruffo Codecasa, Daniela Maria Cirillo, Reinout van Crevel, Dyshelly Nurkartika Pascapurnama, Lidya Chaidir, Stefan Niemann, Roland Diel, Shaheed Vally Omar, Louis Grandjean, Sakib Rokadiya, Arturo Torres Ortiz, Nguyễn Hữu Lân, Đặng Thị Minh Hà, E. Grace Smith, Esther Robinson, Martin Dedicoat, Le Thanh Hoang Nhat, Guy E. Thwaites, Le Hong Van, Nguyen Thuy Thuong Thuong, Timothy M. Walker.

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
