## [Decision Letter · Decision Letter 0]

30 Jun 2023

PGPH-D-23-00405

The effect of M. tuberculosis lineage on clinical phenotype: a retrospective observational study

Dear Dr. Walker,

Thank you for submitting your manuscript to PLOS Global Public Health. After careful consideration, we feel that it has merit but does not fully meet PLOS Global Public Health’s publication criteria as it currently stands. Therefore, we invite you to submit a revised version of the manuscript that addresses the points raised during the review process.

After several rounds securing reviewers for your manuscript, we finally completed this process. My apologies on this delay. 

Before your manuscript can be formally accepted you will need to complete some formatting changes, summarized here: 

Introduction:

81. The objective of the study is well described, as well as the analytical framework necessary for the data and variables of interest.

Methods:

109. Reference is made to the use of the Timika Score for cavitations in chest X-rays. I recommend including an analysis of the disadvantages of this score and citing it. For example, in reference 14 punctuation is described but its disadvantages are not considered. Might consider: Chakraborthy A, Shivananjaiah AJ, Ramaswamy S, Chikkavenkatappa N. Chest X-ray score (Timika score): a useful adjunct in predicting treatment outcome in tuberculosis. Adv Respir Med. 2018;86(5):205-210. doi: 10.5603/ARM.2018.0032. PMID: 30378646.

The statistical analyzes look adequate and consider avoiding bias in the interpretation of the results. Especially when considering the role of lineage 1 in the extrapulmonary manifestations of TB and the greater transmissibility associated with pulmonary TB in the other lineages.

Results:

204. As mentioned in the introduction, there are other TB lineages identified, however, for reasons also explained by the authors in this section, these lineages have not been included because they are very rare or poorly represented, however, for didactic reasons I recommend naming them and location.

Discussion:

423. In this section the authors make some interesting conclusions, especially the possibility that lineages 2, 3 and 4 are associated with pulmonary manifestations and therefore with greater transmissibility. I would recommend placing some reference or including a sentence regarding this important fact, especially considering that in other Latin American countries where there is transmission and circulation of MDR-TB, such as Haiti and the Dominican Republic, this could open the possibility of future phylogenetic analysis. of TB and how TB control programs can focus their strategies.

In general, I congratulate the authors for this work, I hope they can do another work where the genomic characteristics of this lineage and its role in the expression of the disease can be analyzed in depth.

In addition, please take a look at reviewers´s notes below. 

Please submit your revised manuscript by . If you will need more time than this to complete your revisions, please reply to this message or contact the journal office at globalpubhealth@plos.org. Please include the following items when submitting your revised manuscript:

We look forward to receiving your revised manuscript.

Kind regards,

Marianne Clemence, Staff Editor, on behalf of,

Robert Paulino-Ramirez

Staff Editor

Journal Requirements:

2. Please update your online Competing Interests statement. If you have no competing interests to declare, please state: “The authors have declared that no competing interests exist.”

3. Please provide separate figure files in .tif or .eps format only and ensure that all files are under our size limit of 10MB.

4. Please ensure that the Title in your manuscript file and the Title provided in your online submission form are the same.

5. We have noticed that you have uploaded Supporting Information files, but you have not included a list of legends. Please add a full list of legends for your Supporting Information files after the references list.

6. We notice that your supplementary figures are uploaded with the file type 'Figure'. Please amend the file type to 'Supporting Information'. Please ensure that each Supporting Information file has a legend listed in the manuscript after the references list.

Additional Editor Comments (if provided):

Reviewers' comments:

Reviewer's Responses to Questions

**Comments to the Author**

1. Does this manuscript meet PLOS Global Public Health’s publication criteria? Is the manuscript technically sound, and do the data support the conclusions? The manuscript must describe methodologically and ethically rigorous research with conclusions that are appropriately drawn based on the data presented.

Reviewer #1: Yes

Reviewer #2: Yes

2. Has the statistical analysis been performed appropriately and rigorously?

Reviewer #1: Yes

Reviewer #2: Yes

3. Have the authors made all data underlying the findings in their manuscript fully available (please refer to the Data Availability Statement at the start of the manuscript PDF file)?

Reviewer #1: Yes

Reviewer #2: Yes

4. Is the manuscript presented in an intelligible fashion and written in standard English?

Reviewer #1: Yes

Reviewer #2: Yes

5. Review Comments to the Author

Reviewer #1: Interesting paper reinforcing the evidence on lineage effects impacting clinical phenotypes. I appreciate the efforts made to explain the rationale behind the pooling of other existing study datasets, including previously published studies from the same authors (e.g. ref 23) that did not investigate this particular manuscript's hypotheses.

Although using very diverse and not necessarily comparable datasets at first, the authors appropriately tailored their analyses to extract relevant and statistically valid interpretations, and clearly delineated the limitations of their approach and conclusions.

I have no hesitation to recommend this manuscript for publication. I have a very minor detail to raise, not that it should impact the recommendation: the abstract refers to 8 lineages of MTB sensu stricto but I'm only aware of 6 lineages (1, 2, 3, 4, 7, 8), although the authors might be more in-the-know as to the latest lineages discovered in the recent year.

Reviewer #2: Grateful for the opportunity to review the manuscript "The effect of M. tuberculosis lineage on clinical phenotype: a retrospective observational study" (manuscript number PGPH-D-23-00405 ).

The study was well designed, and the findings were adequately presented and discussed, including the authors doing very well to acknowledge and carefully describe the limitations of the study.

In general, studies on the association between M. tuberculosis strain and clinical phenotype are important and should be treated with caution. Congratulations to the authors for the excellent work!

I recommend publishing this manuscript in PLOS Global Public Health.

6. PLOS authors have the option to publish the peer review history of their article (what does this mean?). If published, this will include your full peer review and any attached files.

**Do you want your identity to be public for this peer review?** For information about this choice, including consent withdrawal, please see our Privacy Policy.

Reviewer #1: **Yes: **Antoine Corbeil

Reviewer #2: No

---

## [Decision Letter · Decision Letter 1]

31 Oct 2023

PGPH-D-23-00405R1

The effect of M. tuberculosis lineage on clinical phenotype

Dear Dr. Walker,

Thank you for submitting your manuscript to PLOS Global Public Health. After careful consideration, we feel that it has merit but does not fully meet PLOS Global Public Health’s publication criteria as it currently stands. Therefore, we invite you to submit a revised version of the manuscript that addresses the points raised during the review process.

We look forward to receiving your revised manuscript.

Kind regards,

Mareli Misha Claassens

Guest Editor

Journal Requirements:

Additional Editor Comments (if provided):

Reviewers' comments:

Reviewer's Responses to Questions

**Comments to the Author**

1. If the authors have adequately addressed your comments raised in a previous round of review and you feel that this manuscript is now acceptable for publication, you may indicate that here to bypass the “Comments to the Author” section, enter your conflict of interest statement in the “Confidential to Editor” section, and submit your "Accept" recommendation.

Reviewer #1: All comments have been addressed

Reviewer #3: All comments have been addressed

Reviewer #4: (No Response)

2. Does this manuscript meet PLOS Global Public Health’s publication criteria? Is the manuscript technically sound, and do the data support the conclusions? The manuscript must describe methodologically and ethically rigorous research with conclusions that are appropriately drawn based on the data presented.

Reviewer #1: Yes

Reviewer #3: Yes

Reviewer #4: Yes

3. Has the statistical analysis been performed appropriately and rigorously?

Reviewer #1: Yes

Reviewer #3: Yes

Reviewer #4: Yes

4. Have the authors made all data underlying the findings in their manuscript fully available (please refer to the Data Availability Statement at the start of the manuscript PDF file)?

Reviewer #1: Yes

Reviewer #3: Yes

Reviewer #4: Yes

5. Is the manuscript presented in an intelligible fashion and written in standard English?

Reviewer #1: Yes

Reviewer #3: Yes

Reviewer #4: Yes

6. Review Comments to the Author

Reviewer #1: No additional comments to provide, all original comments were addressed.

Reviewer #3: (No Response)

Reviewer #4: Review of Walker et al, The effect of M. tuberculosis lineage on clinical phenotype

The authors conduct a retrospective observational analysis pooling international datasets to quantify the effect of M. tuberculosis lineage on clinical disease presentation. I congratulate the authors on this important piece of work and appreciate the causal inference approach they took. Pooling observational data from diverse settings and countries is challenging and the authors did well in voicing the limitations of this approach.

I have a few comments below to improve the manuscript:

1. The authors could comment on the advantages of using causal mediation analysis to quantify effects of mediators as opposed to multivariate models.

2. Second generation immigration, i.e. where country of birth is UK or Germany but likely frequent travel to the parents’ country of birth, could also be an important confounder.

3. Please use larger fonts in figures

4. Briefly describe on your imputation approach in the methods section

5. spelling error L288: that ‘it’ might behave similar

6. ‘TB osteomyelitis was significantly more likely than non-meningeal foci of extra-pulmonary TB (“other”) for lineage 1 as compared to lineages 2, 3 and 4 (p=0.032, p=0.01, and p=0.049 respectively. This sentence is unclear since TB osteomyelitis is a non-meningeal focus?

7. These intrinsic biases need to be recognised when constructing each of the presented models. However, none of the studies we pooled data from selected on the basis of lineage as such data could not be known until after the isolates had been analysed. As such each collection had the opportunity to inform on the distribution of lineages within it.

The authors adequately described the limitations of pooling data from different studies for their purpose, i.e., by presenting effect estimates from country specific models. However, I think their last statement above is incorrect. The different selected studies, i.e. those studying drug resistant or meningeal TB, will be biased towards lineages if the authors assumption is that underlying lineages differ in resistance profiles and disease presentation.

7. PLOS authors have the option to publish the peer review history of their article (what does this mean?). If published, this will include your full peer review and any attached files.

**Do you want your identity to be public for this peer review?** For information about this choice, including consent withdrawal, please see our Privacy Policy.

Reviewer #1: **Yes: **Antoine Corbeil

Reviewer #3: **Yes: **Bouke de Jong

Reviewer #4: **Yes: **Matthias Gröschel

---

## [Editor Report · Decision Letter 2]

17 Nov 2023

The effect of M. tuberculosis lineage on clinical phenotype

PGPH-D-23-00405R2

Dear Dr. Walker,

We are pleased to inform you that your manuscript 'The effect of M. tuberculosis lineage on clinical phenotype' has been provisionally accepted for publication in PLOS Global Public Health.

Best regards,

Mareli Misha Claassens

Guest Editor